# An analysis of pacing profiles in sprint kayak racing using functional principal components and hidden Markov models

Harry Estreich[1,2]*, Nicola Bullock[3,4,5], Mark Osborne[3], Edgar Santos-Fernandez[1,2], Paul Pao-Yen Wu[1,2]

**1** School of Mathematical Sciences, Queensland University of Technology, Brisbane, Queensland, Australia, **2** Centre for Data Science, Queensland University of Technology, Brisbane, Queensland, Australia, **3** Paddle Australia, Gold Coast, Queensland, Australia, **4** Australian Institute of Sport, Gold Coast, Queensland, Australia, **5** Bond Institute of Health and Sport, Bond University, Gold Coast, Queensland, Australia

☯ These authors contributed equally to this work.

\* harry.estreich@hdr.qut.edu.au

## Abstract

This study analysed sprint kayak pacing profiles in order to categorise and compare an athlete's race profile throughout their career. We used functional principal component analysis of normalised velocity data for 500m and 1000m races to quantify pacing. The first four principal components explained 90.77% of the variation over 500m and 78.80% over 1000m. These principal components were then associated with unique pacing characteristics with the first component defined as a dropoff in velocity and the second component defined as a kick. All other defined characteristics were a variation of these two, i.e., late kick. We then applied a Hidden Markov model to categorise each profile over an athlete's career, using the PC scores, into different types of race profiles. This model included age and event type and identified a trend for a higher dropoff in development pathway athletes. Using the four different race profile types, four athletes had all their race profiles throughout their careers analysed. It was identified that an athlete's pacing profile changes throughout their career as an athlete matures. This information provides coaches, practitioners and athletes with expectations as to how pacing profiles change across the course of an athlete's career.

## Introduction

Predictive models of individual athlete pacing in competitive kayak races and their change over a career can be used by coaches and sports scientists to better understand athlete progression and optimise strategies for peak performance. The pacing profile refers to the actual differences or changes in speed throughout a race whereas a pacing strategy is the planned changes in speed over the race. Three

**Data availability statement:** A de-identified data set has been uploaded to GitHub: https://github.com/harryestreich1/pacingprofilesanalysis.

**Funding:** The author(s) received no specific funding for this work.

**Competing interests:** The authors have declared that no competing interests exist.

main pacing strategies have emerged in athletic competition based off analysing split times: (i) negative pacing, where speed increases over the course of the event, (ii) positive pacing, where speed decreases, and (iii) even pacing, where speed remains similar over an event [1].

However, when higher resolution timing data is used with more frequent splits, additional pacing profiles were observed [1]. These include an all-out pacing where athletes quickly accelerate to their maximum speed before a continuous decrease in speed throughout the rest of the event. Parabolic-shaped race profiles have three forms: (i) U-Shaped profiles have the slowest section in the middle of the event, (ii) J-Shaped profiles have the slowest section at the start before a gradual increase in pace, and (iii) reverse J-Shaped profiles have a gradual decrease in speed before an increase in speed before the finish. A similar type of profile is a Seahorse-shaped pacing strategy where athletes increase pace near the finish before dropping off before the end [2]. Lastly, a variable pacing profile is when an athlete has a distinct change in pace at certain points often due to external factors such as weather events or changes.

Typically, the sprint kayak pacing profile and strategy tends to vary by distance (200m, 500m, 1000m) and event (K1, K2, K4). Over short distance events (e.g., 200m), all out or positive pacing profiles were evident in para-canoe [3] and able-bodied kayaking [2]. Positive profiles were evident in 500m; however, this differed over the longer 1000m event. For the 1000m events, two different pacing strategies have been identified; a seahorse-shaped profile [2], and a reverse j-shaped profile [4]. The reverse j-shaped profile is also a dominant profile in rowing, who compete over 2000m [5]. Both of these papers describing pacing profiles in sprint kayak used split compared split times and other method to compare them was to use ANOVAs to determine whether there is a statistically significant average speed difference between splits [2]. Another method for identifying pacing profiles, used in swimming [6], was to calculate the slope of a linear regression line for certain sections of a race and then defining each profile as positive, even or negative pacing using a set criteria. Additionally, these studies normalised their data sets due to competition occurring in an outdoor environment, with varying water temperature, water salinity, and weather conditions. Normalising the data is required in order to compare across different races with varying environmental conditions and water conditions. However, none of the reviewed approaches considered the longitudinal evolution of pacing profiles over individual athlete careers. In addition, the use of the different and infrequent time splits imposes an artificial interpretation of race profile velocities, which can lead to differing interpretations of the same race.

Modelling of pacing profiles and their evolution over an athlete's career can not only support targeted interventions for coaching, but also help identify groups of similar athletes for mutual learning. Additionally, a longitudinal model of athlete's pacing profile throughout their career enables the incorporation of additional explanatory factors including age (U18, U21, U23, Open) and event significance (Domestic competitions, World Cups, World Championships/Olympics) and uncertainty in their effects on the profiles. The effect of these explanatory factors has been explored in previous

sports-based literature. A study identified that older athletes had a more even pacing profile compared to younger athletes in marathon running [7] and another study observed a difference in pacing pattern depending on race type [8]. Therefore analysing both the age and event significance of each race profile will be a key focus of this paper.

Statistical models such as principal component analysis (PCA) have been identified as an alternative technique for analysing pacing profiles. PCA is a dimension reduction technique that can be utilised to quantify the variation in split times for each race profile [9]. One approach to classify race profiles that is more robust to difference in split time resolution is to use a curve to approximate the race profile (speed over time). Known as functional Principal Components Analysis (fPCA), this approach has been used to identify the statistically significant variation between the top 3 and bottom 3 athletes in 1000m finals at international sprint kayak competitions based on normalised velocity data [10]. fPCA fits curves to discrete split times and transforms these curves onto principal component (PCs), which are also curves. The algorithm fits PCs with the goal of explaining the majority of the variation in the data using the first few PCs, typically sorted in decreasing order of variance explained. As a result, each PC can potentially map to key pacing profile characteristics, derived organically from the data (real-world race performances). As an extension to fPCA, several studies used cluster analysis to categorise different progression curves in swimming [11] and to categorise different positions in rugby league [12]. These studies used a variety of two-step and model-based clustering techniques to categorise their data sets. It was identified that the inclusion of key variables, including event type and race type, within the clustering process was necessary in order to identify the effects of certain variables. Additionally, a key consideration with the model is the ability to model an athlete's trends throughout their career.

Hidden Markov modelling (HMM) has widespread use in sports data analysis and has been identified as an alternative to clustering to explicitly captures changes in cluster, referred to as a state in this framework, over time. HMMs have been used to model different levels of control in football [13] and infer stroke phase in swimming [14]. Additionally, HMMs have also been used to model longitudinal data and to infer an unobserved state that is changing over time. For example, in basketball [15] it was used to infer athlete streakiness (streaky or non-streaky performance) and in baseball [16] it was used to predict future home run totals. Noticeably, the paper used a vector of covariates (home ballpark and position) within the HMM framework to capture different impacts of these covariates with state. This concept is applicable within the HMM of this paper as several variables (e.g., age, race type) influence the type of pacing profile. HMMs have also been used to quantify different types of physical activity using accelerometer data [17]. This concept of classifying data points into different categories is directly applicable in this paper as this paper's goal is to identify and classify different types of race profiles. Therefore, HMMs will be explored as a potential method for modelling the evolution of an athlete's pacing profile.

The aim of this study was to derive categories of pacing profiles and track their change over time for individual athletes. There are two parts to this: (i) using fPCA to derive key patterns in sprint kayak pacing profiles using the data, and (ii) HMM modelling clusters (or states) of pacing profiles and their change over time for individual athletes, given covariates of age and event, capturing the uncertainty inherent in race performances. The latter will allow for the analysis and prediction of transitions in pacing profile for individual athletes.

## Methods

### Data

Race performance data was collected across a large set of athletes in domestic (National domestic regattas, National Championships and Selection Regattas) and international competitions (World Cups, World Championships and Olympic Games, Junior World Championships, U23 World Championships) between 2010 and 2023. The data set included racing across two different races, Women's K1 500m and Men's K1 1000m. Within each data set, some athletes were included even though they only had a few races within the data set. These athletes were kept in the data set in order to include more expansive and diverse race types however only athletes who have a significant number of races only a long period

of time were analysed individually. Additionally, although the initial data set included different event phases including heats, semi-finals and finals, heat races from domestic competitions were removed from the data set as it was determined that they were often not a true reflection of an athlete's pacing profile due to a lack of competitive depth in some of these events. Outlier removal was conducted based on input from domain experts, who leveraged their deep understanding of the data-generating process to identify observations that were inconsistent with the expected patterns or operational constraints of the system. This expert-driven approach ensured that the outliers identified were truly anomalous and not representative of the underlying phenomena being studied

In the Women's K1 500m data set, there were 70 athletes across a variety of age groups (U18, U21, U23, Open) and event phases (heat, semi-finals, final). Each race is classified as either a domestic event, non-Championship international race (World Cup, Junior and U23 World Championships) or a Championship international race (World Championships, Olympics). Of the 70 athletes, 13 athletes had at least 10 races and 9 athletes had more than 30 races with two athletes, Athlete A (90 races) and Athlete B (75 races) having the most. The Men's K1 1000m data set had 67 athletes with 18 athletes having at least 10 races. Two athletes, Athlete C (59 races) and Athlete D (32 races) where identified as having a large number of races across many levels of competition and multiple age groups.

From each race, pacing profiles were created from athlete velocity data which included interval times for each 50m segment. The initial unnormalized data for the Women's K1 500m had an average velocity of 4.29m/s (SD 0.23m/s) and an average maximum velocity of 4.76m/s (SD 0.29m/s) whilst the Men's K1 1000m had an average of velocity of 4.8m/s (0.24 m/s) and an average maximum velocity of 5.49m/s (SD 0.3m/s). This data set was then normalised using the average velocity for the entire race. Therefore, each 500m race profile had 10 data points describing an athlete's velocity and each 1000m race had 20 data points. As each race was conducted under different environmental conditions it is necessary to normalise the velocity data by average boat speed for each race profile. By normalising the data, the athlete's overall boat speed throughout the race is no longer comparable and it is the athlete's strategy or pacing performance that is analysed. In Fig 1, the mean pacing profile for the Women's K1 500m shows that athletes tend to increase their velocity until reaching their peak segment velocity at approximately 80m before continuously decreasing through to the end. In Fig 2, the mean pacing profile for Men's K1 1000m shows that athletes tend to increase speed and reach their peak velocity at approximately 100m into the race before decreasing in speed until the 500m mark. For the second half of the race, athletes tend to maintain most of their speed except for a small kick in the last 250m of the race. This data was collected

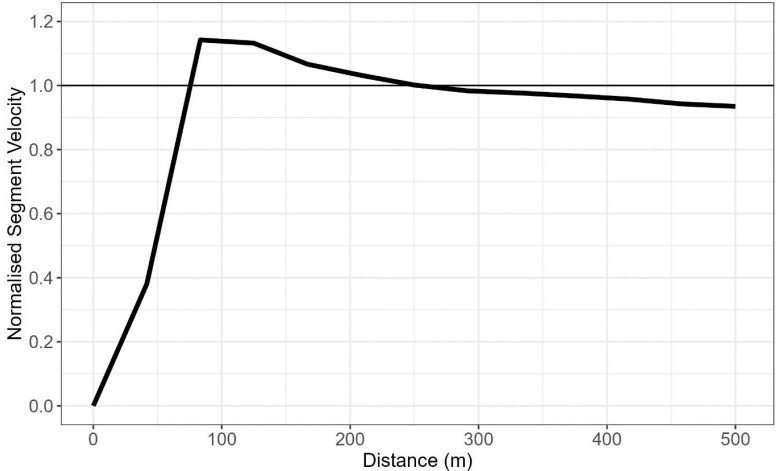

**Fig 1. Mean Pacing Profile for the Women's K1 500m data set.** The mean normalised segment velocity for each split time is calculated from all athletes within the data set.

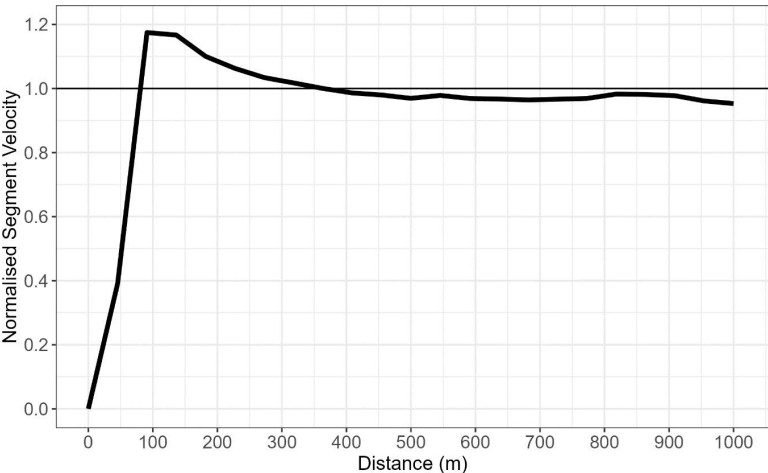

**Fig 2. Mean Pacing Profile for the Men's K1 1000m data set.**

using a 10 Hz GPS device (Catapult Sport, Melbourne, Australia) with each 50m split identified and analysed for the change in time which was used to calculate all velocity for each segment.

## Statistical methods

The statistical methods part is split into two sections. The first section involves fPCA and quantifies the variation in each data set using principal components (PCs) and the resulting principal component scores can be used to describe each race profile in terms of each principal component. The computation in this section was done using R [18], specifically the fda package [19]. The second section uses the PC scores calculated in the first section to globally-cluster different types of race profiles using a hidden Markov model (HMM), using the depmixS4 package [21]. Both sections were completed independently for both data sets.

## Functional PCA

In our framework, the discrete data set of split times must first be transformed into functional data [20]. To do this each of the pacing profiles are represented using a b-spline basis and weighted eigenfunction, calculated from the discrete segment split times. Firstly, let $x$ be the distance along the race in metres and $f(x)$ is the smoothed version of the normalised race velocity calculated using the b-splines basis. The set of principal component scores for the first PC, $\beta_1$, is computed from using:

$$\beta_1 = \int_{x_1}^{x_p} \Phi_1(x) f(x) dx \tag{1}$$

where $\Phi_1(x)$ is the first principal component, $f(x) \in [f_1(x), \ldots, f_n(x)]$), is the set of smoothed pacing profiles. $x_1$ is 0, representing the start of the race in metres, and $x_p$ is 500 or 1000, the end of race in metres (this differs for the two different data sets). Each principal component is calculated by maximising the variance of the set of first principal component scores:

$$\left\| \Phi^2_1(x) \right\| = \int_{x_1}^{x_p} \Phi^2_1(x) dx \tag{2}$$

Each successive principal component is then calculated using the same methods however each principal component is orthogonal to the previous one. The percentage of variation described by each principal component can be calculated and generally a significant amount of variation can be described by the first few principal components.

Each of these principal components, also known as eigenfunctions, map to the data set and therefore each principal component can be analysed to determine what characteristics of a pacing profile are evident within each eigenfunction. Therefore, from each pacing profile the PC scores can be used to indicate how each principal component describes each profile with each fPC (functional Principal Component) score calculated from a linear combination of the eigenfunction and the mean-centred pacing profile. Therefore, a more positive fPC score will often indicate that the mean-centred pacing profile is more similar to the eigenfunction.

The application of fPCA to the two data sets produced four principal components in each data set that describe a significant amount of variation. In the Women's K1 500m data set, PC1 and PC2 explaining a majority of the variation with 75.58% accounted for with the first two components, with the addition of PC3 and PC4, 90.77% of the variation was explained using the first four principal components. In the Men's K1 1000m data set, 62.88% of the variation was explained with PC1 and PC2 and 78.8% explained by the first four principal components. In this case, the first four PCs can explain the vast majority of the variation in the data set. Additionally, utilising four instead of two PCs improved the practical interpretation of the model by capturing more distinct pacing profiles. Therefore, by using fPCA each pacing profile can be reduced to a few fPC scores that still account for the majority of the variation in the data set. These fPC scores can be analysed to categorise different types of profiles using a HMM which can be used to analyse pacing profiles trajectories over a career.

## Hidden Markov model

The hidden Markov model is built using a set of known observations, the PC scores and explanatory variables, gender and event type, and a set of unknown hidden states, which represent clusters of pacing profiles with similar characteristics (see Discussion for characteristics derived from principal components). Each state describes a certain type of pacing profile how an athlete's career trends between these different states can be used to identify the changes in pacing profile throughout an athlete's career.

As there are multiple observed variables a multi-variate hidden Markov model is necessary and the model is also set up such that each athlete is its own independent continuous time-based stochastic process. It was also determined that a four-state model was the most appropriate, as the AIC (Akaike Information Criteria) value, an estimator of model fit and prediction error, did not improve by increasing the number of states beyond 4 (see Appendix 1 in S1 File). Additionally, as the starting values can change the results of the hidden Markov model, the best performing model was found by 200 model repeats to find the model with the lowest log-likelihood. Therefore only the first four principal components were used to model the response (observations). There are two main sets of parameters that are fitted using the data sets [21]. Firstly, there is a constant transition matrix, which is an 4x4 matrix (given it is a four-state model), which contains the transition probabilities between each state as well as an initial state probability vector, these are both independent of athletes. This can be modelled using the joint likelihood of observations $\boldsymbol{O}_{1:T}$, which is a multivariate normal distribution with a 4x1 mean vector and 4x4 covariate matrix, and latent states $\boldsymbol{S}_{1:T}$, given model parameters $\theta$ and covariates $\boldsymbol{z}_{1:T} = (\boldsymbol{z}_1, \ldots, \boldsymbol{z}_t)$ [22]

$$P(\boldsymbol{O}_{1:T}, \boldsymbol{S}_{1:T}|\theta, \boldsymbol{z}_{1:T}) = \pi_1 \boldsymbol{b}_{S_t}(\boldsymbol{O}_1|\boldsymbol{z}_1) \prod_{t=1}^{T-1} a_{ij} \boldsymbol{b}_{S_t}(\boldsymbol{O}_{t+1}|\boldsymbol{z}_{t+1}) \tag{3}$$

Where $S_t$ is an element of S = {1 … n}, a set of n latent states. $\pi_1 = P(S_1)$ is a vector of the initial probabilities. $a_{ij} = P(S_{t+1} = j|S_t = i)$ provides the probability of a transition from state $i$ to state $j$. $\boldsymbol{b}_{S_t}$ is a vector of observation densities $\boldsymbol{b}^k{}_{S_t} = P(O^k{}_t|S_t = j, t)$ that provides the conditional densities of observations $O^k{}_t$ associated with latent state $j$ and

covariate $z_t$. In this hidden Markov model these observation densities were modelled using Gaussian distributions. This was verified by checking the distributions of each principal component using fitdistrplus package [22] (see Appendix 3 in S1 Fig). Each Gaussian distribution also has a state-specific variance (see Appendix 2 in S1 File). Residual analysis also confirmed that the model is suitable for the data (see Appendix 5 in S1 Fig). Each of these parameters have been estimated using the depmixs4 package [23], this package uses an estimation-maximisation algorithm which iteratively maximising the expected joint log-likelihood of the parameters. Additionally, the inferred state for each data point is obtained through global decoding.

Additionally, for each state, S1 to S4, there is a linear model for the emission probabilities relating the unobserved latent state, i.e., pacing profile pattern cluster, to the observed PC score variables, i.e., pacing profile pattern. Each linear model takes into account covariates of event type and age group. Each of these approaches are modelled using a multivariate regression equation (Equation 4) where the mean response corresponding to PC1–4 are $\{\mu_1, \mu_2, \mu_3, \mu_4\}$, $\beta_{ijk}$ is the coefficient for each variable where $i$ = principal component, $j$ = state and $k$ = variable. Additionally, the covariates values (either 0 or 1) are defined as $x_1$ = Age Group U21, $x_2$ = Age Group U23, $x_3$ = World Cup/ Juniors Event, $x_4$ = World Championships/ Olympics. Equation 4 is an example equation for the Women's K1 500m data set with the Men's K1 1000m HMM having an additional variable for Age Group U18.

$$\mu_1 = P(S_1) * (\beta_{110} + \beta_{111} * x_1 + \beta_{112} * x_2 + \beta_{113} * x_3 + \beta_{114} * x_4) + \ldots + P(S_4) *$$
$$(\beta_{140} + \beta_{141} * x_1 + \beta_{142} * x_2 + \beta_{143} * x_3 + \beta_{144} * x_4) + \ldots \tag{4}$$

## Results

### fPCA

**fPCA Eigenfunctions.** The fPCA model was fit to both data sets with the principal component eigenfunctions, PC1 though PC4, are shown in Figs 3 and 4. As principal component scores are a linear combination of each eigenfunction and each pacing profile relative to the mean pacing profile, a pacing profile whose relative segment velocities have the same sign as the eigenfunction shown will have more positive PC scores. Additionally, higher absolute eigenfunction values will have a larger effect on PC scores and therefore most important when identifying how a particular principal component correlates to in the data set.

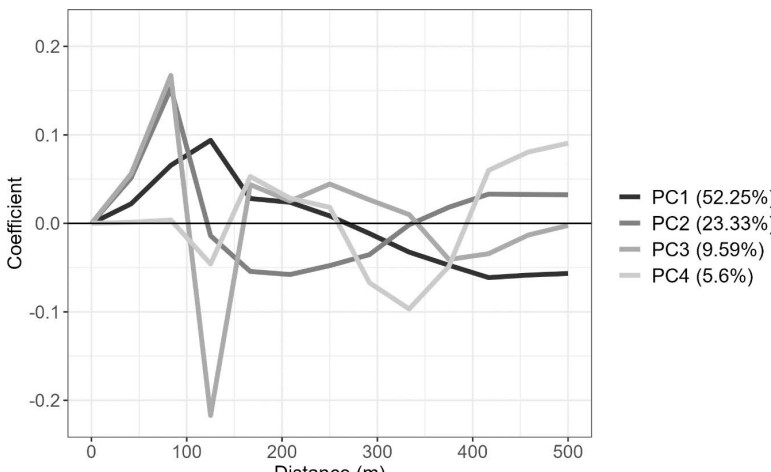

**Fig 3. The first four principal components for the Women's K1 500m race profile data set, plotted against distance.**

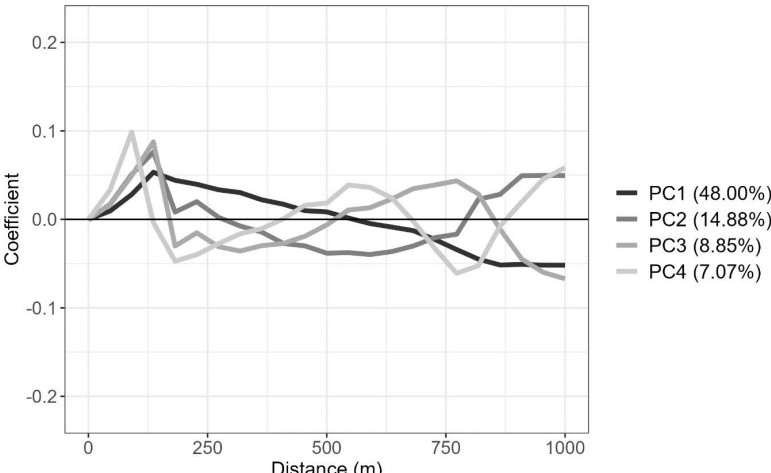

**Fig 4. The first four principal component eigenfunctions for the Men's K1 1000m race profile data set, plotted against distance.**

In Fig 3, each principal component in the Women's K1 500m can be analysed to determine what each PC score indicates. The PC1 eigenfunction indicates that a positive PC1 score corresponds to an above average peak normalised segment velocity in the first half of the race, peaking at approximately 125m, and a large contrast between the peak and minimum segment velocity. Therefore the particular pacing characteristic that PC1 corresponds with is defined as dropoff. A positive PC2 score indicates a velocity profile where an athlete velocity dips between 100m and 250m and begins increasing to above average velocity from the 300m mark, this is defined as kick. A positive PC3 score corresponds to a sharp contrast between the peak at approximately 80m and a low trough at approximately 120m before sinusoidal trends for the rest of the race, this is defined as early dropoff. A positive PC4 corresponds to a significant drop in speed between the 200m and 300m mark before significantly increasing for the last 200m mark, this is defined as a late kick.

In Fig 4, the PC1 eigenfunction is similar to the PC1 eigenfunction in Fig 3 as a positive PC1 score indicates above average normalised segment velocity in the fast half of the race before consistently decreasing through the rest of the race. PC2 is again similar, however, the minimum point in the eigenfunction is later in the race at approximately the 600m mark. PC3 indicates a consistent increase in speed between the 250m and 750m mark before a dropoff in speed in the last 250m. This is considerable different to the Women's K1 500m data set and therefore a new definition, late dropoff, it used for this data set. PC4 has a significant dropoff at the 500m and a kick to end the race. PC1, PC2, and PC4 have similar characteristics across both data sets are therefore the definitions are the same across both data sets.

### Hidden Markov model

The PC scores were fit to a HMM and a summary of the results are shown in Tables 1–4. Table 1 contains the intercept coefficients for the emission equation for each state and principal coefficient combination. These values represent the centroid for each state when plotted in 4-dimensional space when using baseline variables. These baseline variables are domestic event type and Open age group. These intercept values can be used to determine the mean race profile characteristics for each state as described by the PC scores. Therefore, higher PC scores indicate a state is more likely to exhibit the particular characteristics of the principal component.

The connection between each state and the principal component scores are shown in Fig 5. This diagram also utilises the definitions for each principal component that are defined later in the Discussion to allow for the easy connection between HMM state, principal component and pacing characteristic. For example, in the Women's K1 500m dataset, State

**Table 1. Intercept coefficients for each distribution describing the mean PC scores of each state (row) and principal component (column) for Women's K1 500m and Men's K1 1000m. These values indicate that centroid for each state when using the baseline variables.**

| Women's K1 500m | | | | | Men's K1 1000m | | | |
|---|---|---|---|---|---|---|---|---|
| | PC1 | PC2 | PC3 | PC4 | PC1 | PC2 | PC3 | PC4 |
| State 1 | 0.173 | 0.230 | −0.003 | −0.028 | −0.871 | 0.182 | −0.126 | −0.137 |
| State 2 | −0.418 | 0.552 | −0.136 | 0.168 | 0.979 | 0.287 | 0.030 | −0.068 |
| State 3 | −0.373 | −0.105 | 0.070 | 0.050 | −0.107 | 0.654 | 0.080 | 0.185 |
| State 4 | 0.063 | −0.045 | −0.045 | −0.039 | 0.118 | 0.007 | −0.023 | 0.030 |

**Table 2. Effect of event type on emission probability coefficients obtained from the HMM emission equation (equation 4). Rows are state and columns are each combination of principal component (PC1 through PC4) and event type (Domestic (baseline value), World Cup/ Juniors, World Championships/ Olympics). Each event type coefficient is relative the baseline Domestic event and a more positive coefficient indicates a higher probability for the given state.**

| | PC1 | | PC2 | | PC3 | | PC4 | |
|---|---|---|---|---|---|---|---|---|
| | World Cup/ Juniors | World Champs | World Cup/ Juniors | World Champs | World Cup/ Juniors | World Champs | World Cup/ Juniors | World Champs |
| **Women's K1 500m** | | | | | | | | |
| State 1 | −0.570 | 0.045 | −0.273 | −0.176 | 0.190 | 0.262 | −0.016 | 0.073 |
| State 2 | 0.051 | 0.246 | −0.045 | 0.154 | −0.108 | −0.154 | −0.009 | −0.012 |
| State 3 | 0.363 | 0.618 | −0.241 | −0.384 | −0.092 | 0.037 | −0.012 | −0.277 |
| State 4 | 0.078 | 0.060 | −0.051 | −0.047 | 0.018 | 0.048 | 0.014 | −0.010 |
| **Men's K1 1000m** | | | | | | | | |
| State 1 | 0.061 | 0.023 | −0.088 | −0.119 | 0.088 | 0.120 | 0.084 | 0.062 |
| State 2 | 0.585 | 1.355 | 0.285 | −0.338 | 0.104 | 0.647 | −0.664 | −0.422 |
| State 3 | 0.134 | 0.310 | −0.493 | 0.404 | −0.193 | −0.547 | −0.225 | 0.174 |
| State 4 | 0.060 | 0.424 | −0.082 | 0.065 | −0.042 | −0.062 | −0.0.96 | −0.047 |

**Table 3. Effect of age group on emission probability coefficients obtained from the HMM emission equation (equation 4). Rows are state and columns are each combination of principal component (PC1 through PC4) and age group (Open (baseline coefficient), U18, U21, U23). Each event type coefficient is relative the baseline Open age group and a more positive coefficient indicates a higher probability for the given state.**

| | PC1 | | PC2 | | PC3 | | PC4 | |
|---|---|---|---|---|---|---|---|---|
| **Women's K1 500m** | | | | | | | | |
| | U21 | U23 | U21 | U23 | U21 | U23 | U21 | U23 |
| State 1 | −0.016 | −0.024 | −0.098 | −0.040 | 0.028 | −0.055 | 0.100 | 0.040 |
| State 2 | 0.287 | −0.384 | 0.339 | 0.103 | −0.041 | −0.277 | −0.160 | −0.148 |
| State 3 | 0.206 | 0.288 | −0.645 | 0.200 | 0.243 | 0.376 | −0.176 | −0.280 |
| State 4 | −0.383 | −0.136 | 0.090 | 0.103 | −0.023 | −0.010 | 0.041 | −0.021 |

| | PC1 | | | PC2 | | | PC3 | | | PC4 | | |
|---|---|---|---|---|---|---|---|---|---|---|---|---|
| **Men's K1 1000m** | | | | | | | | | | | | |
| | U18 | U21 | U23 | U18 | U21 | U23 | U18 | U21 | U23 | U18 | U21 | U23 |
| State 1 | 1.008 | 0.895 | 1.216 | 0.213 | −0.029 | −0.061 | −0.037 | 0.008 | 0.059 | 0.124 | 0.022 | −0.058 |
| State 2 | −1.710 | −1.656 | −1.529 | −0.291 | −0.410 | −0.238 | −0.022 | 0.238 | 0.234 | 0.326 | 0.312 | −0.049 |
| State 3 | 0.285 | −0.285 | −0.090 | −0.231 | −0.788 | −0.414 | −0.258 | −0.442 | −0.283 | −0.217 | −0.256 | 0.099 |
| State 4 | 0.538 | 0.119 | 0.026 | −0.728 | 0.118 | 0.049 | 0.241 | 0.148 | −0.052 | 0.217 | 0.049 | −0.168 |

**Table 4. The transition matrices for HMMs.**

| Women's K1 500m | | | | | Men's K1 1000m | | | |
|---|---|---|---|---|---|---|---|---|
| | To State 1 | To State 2 | To State 3 | To State 4 | To State 1 | To State 2 | To State 3 | To State 4 |
| **From State 1** | 0.976 | 0 | 0.024 | 0 | 0.805 | 0.081 | 0 | 0.114 |
| **From State 2** | 0 | 0.917 | 0.027 | 0.056 | 0.107 | 0.815 | 0.078 | 0 |
| **From State 3** | 0 | 0.022 | 0.853 | 0.125 | 0.064 | 0.099 | 0.838 | 0 |
| **From State 4** | 0.026 | 0.052 | 0.019 | 0.903 | 0.167 | 0.234 | 0.124 | 0.475 |

1 indicates a positive PC1 and PC2 mean value. Additionally, the thicker the line the larger the absolute value for each PC score is in Table 1.

The model coefficients in Table 2 indicate the effect that each event type variable has on the mean PC scores when in a given state, relative to the baseline category, Domestic. Similarly, the model coefficients in Table 3 indicate the effect that each age group has on the mean PC scores when in a given state, relative to the baseline category, Open. As per Equation 4, the predicted score is a weighted mixture of the state probabilities and coefficient values. Therefore, if the race profile was from a World Championships event in the U23 age group in the Women's K1 500m, using Equation 4, the predicted mean for PC1 would be $\mu_1 = P(S_1) * (0.173 + 0.024 * 1 + 0.045 * 1) + \ldots + P(S_4) * (0.063 + 0.136 * 1 + 0.06 * 1) + \ldots$ with all coefficients retrieved from Tables 1 and 2.

The transition matrices in Table 4 show that transitions between states is possible between all states, however, maintaining the same state between time points is likely for all states. Noticeably, in the Women's K1 500m HMM there is 97.6% probability of remaining in State 1 whereas there is just a 47.5% probability of remaining in State 4 for the Men's K1 1000m HMM.

The HMM output provides a probability that each pacing profile is in a given state. By identifying the most likely state for each pacing profile, as shown in Figs 6 and 7, a career-long trend can be analysed to evaluate how an athlete's pacing profile changes throughout their career. The case study analysis looks into the career-long trends of these four athletes.

## Discussion

The initial findings indicated that for the Women's K1 500m, the average race profile reflected an all-out race strategy where athletes get to their maximum speed as fast as possible and then gradually slowing down as fatigue sets in. In the Men's K1 1000m, the seahorse profile, previously defined was identified in the average race profile as segment velocity began to increase at 750m before dropping throughout the end of race. This suggests that over the longer distance, athletes have the ability to plan their strategy more appropriately by attempting to maintain as much velocity during the middle of the race before increasing speeds at the end of the race. Alternatively, over the shorter 500m the all-out profile is the apparent default strategy as a final acceleration is not viable as maintaining a constant near-maximum pace requires less energy than reaccelerating [1]. This race strategy identification was done visually using Figs 1 and 2 and backs up previously identified strategies [2] and although they help provide an overall interpretation of a race profile, extending to fPCA allows for the key sources of variation in a pacing strategy to be identified.

The principal component eigenfunctions, shown in Figs 3 and 4, have previously been defined using pacing characteristics. It was identified that the pacing characteristics differ between the Women's K1 500m and the Men's K1 1000m, in part due to longer distances requiring different race strategies. The key identified difference is PC3, which is characterised by a dropoff much later in the race in the Men's K1 1000m compared to the Women's K1 500m. A common attribute between both distances for PC2 is the start of the kick, indicated by the increasing eigenfunction curve, is approximately 300m to 400m from the end. This suggest that when an athlete increases their velocity for the kick it occurs a similar

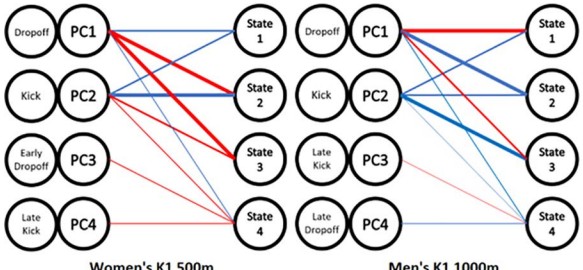

**Fig 5. Diagram showing mean PC scores for the centroid of each state when using baseline variables ( Table 1).** Blue indicating a positive mean, red indicating a negative mean. A thicker line indicates a larger absolute value for each PC score mean. If no line exists, the mean PC scores is significantly smaller than other values for the same state and is therefore not a significant factor in the interpretation.

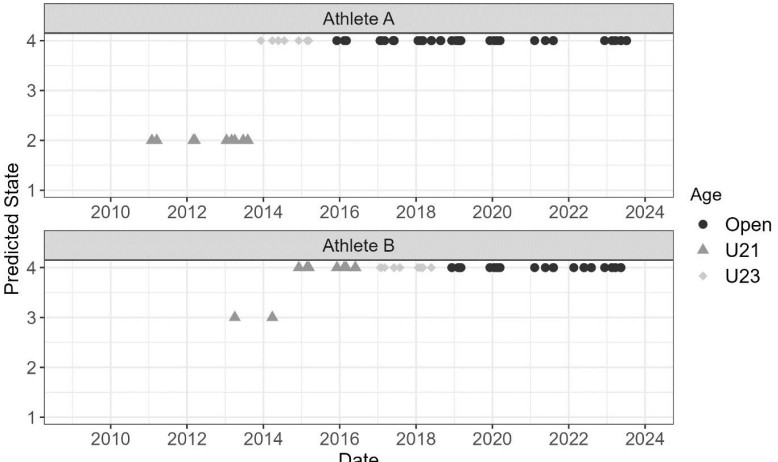

**Fig 6. State for each race profile across Athlete A and B's career.**

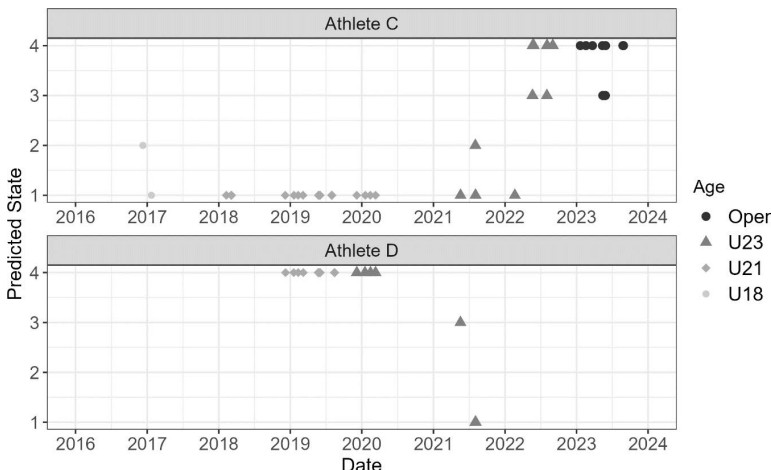

**Fig 7. State for each race profile across Athlete C and D's career.**

distance from the end regardless of the race distance. These identified characteristics allow for coaches and athletes to identify what the core elements of a pacing profile are and provide information on how each athlete competes in terms of these 4 characteristics.

It should be noted that when implementing this two-step fPCA-HMM approach, there might be issues including loss of sensitivity and bias towards low-order PCA components [24], thus there is potential for misleading inference. Several papers have identified methods for including the feature analysis within the framework of an HMM [25–27]. However, the approach of using feature extraction techniques first then subsequently classify using modelling is common as well [28,29]. In this case study, preliminary validation indicated that the proposed model was robust and had been effective in explaining the variation in the data. The RMSE was less than 0.5 for each principal component which was less than the standard deviation of that component. Potentially, fitting both pacing curves and clusters together in a single framework could potentially lead to more robust results with better generalisability and propagate uncertainty [24]. However, as this is less explored with fPCA and HMMs, it is an avenue for future investigation and outside the scope of this study due to the significant increase in complexity and issues with interpretability. Additionally, we calculated the empirical distribution of the sojourn times for each hidden state and compared them to the geometric distribution using goodness-of-fit tests (this was done using the R package fitdistrplus) (see Appendix 4 in S1 Fig). Degeneracy issues were also checked, and there we no collapsed states as the transition matrix (Table 4) allows for transitions out of every state, and emission matrices (Tables 1–3) show that no two states are the same (i.e., not redundant). A potential exploration of this model would be to determine whether a semi-Markov model would be more appropriate, however, the model appears justifiable.

After defining the characteristics that are exhibited in each principal component, the next step is to describe each state from the Hidden Markov model in terms of each characteristic. As shown in Fig 6, the baseline variables from Table 1 can be used to synthesise the relationships between each state and each principal competent. For example, in the Women's K1 500m data set, in State 1, the dominant principal components are both PC1 (0.173) and PC2 (0.230). Therefore State 1 can be defined as indicating high dropoff (PC1) and high kick (PC2) with negligible PC3 and PC4 scores. For States 1, 2 and 3 for both data sets, each state mean centroid is predominantly a combination of PC1 and PC2 scores with PC3 and PC4 have coefficients close to zero which indicates average characteristics. However, State 4, for both data sets, have average scores for all principal components.

The hidden Markov model also has variable coefficients describing the effect of event type (Domestic, World Cup/ Juniors, World Championships/ Olympics) and age group (U18, U21, U23, Open). In the model, these coefficients show the effect that each variable has on the response distribution compared to the baseline variables (Domestic event type and Open age group). Several key trends have been identified in Tables 2 and 3 for the Women's K1 500m data set, these include that PC1 is higher in international events in every state (except for State 1 and World Cup/ Juniors) which indicates that athletes have a higher dropoff in international events. Additionally, PC2 is lower in international events in every state (except for State 2 and World Champs/ Olympics) which indicates that athletes have a lower kick in international events. There are no consistent trends in the age group coefficients, however, for the Women's K1 500m data set, in State 4, PC1 is lower in development age groups and PC2 is higher. This indicates that those in State 4 have lower dropoff and higher kicks in development age groups.

## Longitudinal modelling case study

As defined previously, Athletes A, B, C, D are four athletes who were identified for in-depth analysis due to their significant number of races within the data set. Therefore with the definitions of each principal component and the implications of positive and negative PC scores identified, the calculated PC scores for each athlete can be analysed to identify athlete-specific trends in race strategies. By building two hidden Markov model, one for the Women's K1 500m data set and one for the Men's K1 1000m data set, four identifiable clusters were calculated for each. The state for all four chosen athletes are shown from all races throughout their career in Figs 6 and 7. The state plot can be used to identify and analyse the trends an athlete pacing profile go through throughout their career.

As shown in Fig 6, Athlete A begins their career in State 2, displaying signs of a low dropoff/high kick pacing profile transitioning to State 4, average profile characteristics, when they move from U21s to U23s. Alternatively, Athlete B starts with a few races in State 3, low dropoff/high kick, before transitioning to State 4 for the majority of their career. This appears to indicate that both of these athletes, who have the most pacing profiles in the data set, trends towards an average profile throughout their career.

As shown in Fig 7, Athlete C appears to begin their career in State 1, low dropoff/high kick during U18s and U21s before beginning to shift towards State 4, average profile characteristics, throughout U23 and mostly maintaining State 4 throughout the Open age group. Noticeably, Athlete C is less consistent than Athletes A and B (Fig 6) with several races switching to State 3, low dropoff/low kick, and therefore their race profile appears to be inconsistent. Athlete D differs in that they start from State 4 before transitioning to State 1. However, there is a lack of data beyond this point and whether this change in long term is yet to be seen as the athlete is still early in their open career.

When comparing the trends in the two female athletes compared to the male athletes, the state consistency appears to differ significantly with both female athletes consistently staying in State 4 for the majority of their open careers whereas both male athletes show more inconsistency with both athletes transitioning between states regularly, although they are predominantly staying in State 4. It should be noted that both female athletes have considerably more data points in the Open age group, and they tended to change states throughout the development pathway age categories similar to that observed with the two male athletes. Therefore, it is possible that the male athletes will tend to be more consistent once they have a large data set in the Open age group, as both female athletes did. Lastly, it is clear that all four athletes underwent transitions between different states and this transition often occurred throughout the development pathway for all four athletes although both Athlete A and B were fairly consistent from the U23 age group. Therefore as athletes mature, their physical characteristics change and their pacing profiles change appropriately. This confirms that an athlete's pacing profile can change throughout their career.

## Future work

This model was able to categorise the different types of race strategies into four different states that are described using four different pacing characteristics and this allowed for the career trends of athletes to be analysed. However, there is the opportunity for further analysis to be undertaken to better understand how an athlete race profile changes throughout their career and to potentially forecast how it will change into the future. Firstly, a decision was made to normalise each race profile in order to remove the overall effect of environmental conditions. However, by doing this the overall speed of an athlete, which is likely to change throughout an athlete's career, is no longer able to be considered or evaluated. Therefore, a possible extension of the model is to include race time in order to evaluate whether ability has an effect on an athlete's pacing strategy. Additionally, a key assumption of the modelling is that the environmental conditions are consistent throughout an entire race, which is unlikely to be true and without weather data for every race it is a necessary limitation of the data set. A possible future study could analyse a large data set from one event with accompanying environmental mapping data that can incorporated into the modelling to evaluate the effects of weather on each race profile.

## Practical applications

In this manuscript, several example athletes were analysed using the hidden Markov model and a number of key observations about how their pacing strategy change throughout their career were observed. These include that pacing strategy is not fixed and will change across the course of a sprint kayak athletes' career. Additionally, an athlete was more prone to changes throughout development pathway and appeared to find more consistency once reaching the Open age group. How an athlete's pacing profile changes could provide many benefits to coaches and athletes and the way in which an athlete's strategy affects their performance can be determined. These conclusions could apply to a range of sprint kayak disciplines, multi-athlete disciplines and para events, as well in other sports where pacing is a key component, rowing and

 

swimming. Additionally, the findings identified four different clusters of pacing profiles for both the Women's K1 500m and the Men's K1 1000m which could be used to help determine an athlete's strategy consistency. The model also identified several key trends with respect to age group and event type and knowledge of this will help provide development pathway coaches and athletes with expectations as to how their strategy could or should be expected to change in the future.

## Conclusion

In conclusion, the goal of this manuscript was to identify methods for quantifying and comparing athlete pacing profiles. This was conducted using a combination of fPCA and HMM by identifying four principal components for each data set. Each of these four PCs were then associated with unique pacing characteristics which were than used to categorise each pacing profile into 4 different states using a HMM. Using the 4 HMM states, an athlete's pacing profile can be analysed throughout their career. This analysis concluded that an athlete's pacing profile can change throughout their career and this insight can provide many benefits to coaches and athletes, particularly to development coaches. The longitudinal case study shows the benefits of fPCA and HMM in categorising athletes pacing profiles in sprint kayak.

## Supporting information

**S1 File.** Appendix 1. AIC Values calculated for different number of states in HMM. Appendix 2. Standard Deviation for each state and principal component in the HMM.
(DOCX)

**S1 Fig. Appendix Figures.** Appendix 3. Principal Component distributions for both the Men's and Women's dataset. There is also an example fitdistrplus plot for Men's PC4 validating that the distribution is Geometric. Appendix 4. Histogram of Sojourn times and fitdistrplus plot for both Men's and Women's dataset. Appendix 5. Example histogram of residuals for state 1 in the Men's HMM and a example QQ plot analysis for PC4.
(ZIP)

## Acknowledgments

This research was supported by the Centre for Data Science at QUT and Paddle Australia.

## Author contributions

**Conceptualization:** Nicola Bullock, Mark Osborne, Edgar Santos-Fernandez, Paul Pao-Yen Wu.

**Data curation:** Nicola Bullock, Mark Osborne.

**Formal analysis:** Harry Estreich.

**Investigation:** Harry Estreich.

**Methodology:** Harry Estreich, Edgar Santos-Fernandez, Paul Pao-Yen Wu.

**Supervision:** Nicola Bullock, Mark Osborne, Edgar Santos-Fernandez, Paul Pao-Yen Wu.

**Writing – original draft:** Harry Estreich.

**Writing – review & editing:** Nicola Bullock, Mark Osborne, Edgar Santos-Fernandez, Paul Pao-Yen Wu.

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
