## [Decision Letter · Decision Letter 0]

Thank you for submitting your manuscript to PLOS ONE. After careful consideration, we feel that it has merit but does not fully meet PLOS ONE’s publication criteria as it currently stands. Therefore, we invite you to submit a revised version of the manuscript that addresses the points raised during the review process.

We look forward to receiving your revised manuscript.

Kind regards,

Matteo Vandoni

Academic Editor

PLOS ONE

[N/A]. 

3. In the online submission form, you indicated that [The data is owned by Paddle Australia. If the reviewers require access to the full-dataset as part of the peer review process, a de-identified data set can forwarded upon request.]. 

Additional Editor Comments:

Dear Authors,

The reviewers highlighted several points to improve. Please carefully review the manuscript that is not acceptable in the present form.

Kind regards

Reviewers' comments:

Reviewer's Responses to Questions

**Comments to the Author**

1. Is the manuscript technically sound, and do the data support the conclusions?

Reviewer #1: Partly

Reviewer #2: Partly

2. Has the statistical analysis been performed appropriately and rigorously?

Reviewer #1: No

Reviewer #2: No

3. Have the authors made all data underlying the findings in their manuscript fully available?

Reviewer #1: Yes

Reviewer #2: No

4. Is the manuscript presented in an intelligible fashion and written in standard English?

Reviewer #1: Yes

Reviewer #2: No

Reviewer #1: The manuscript addresses an interesting topic. The data are original and the results could be used to further researches on the topic. The use of the hidden Markov models (HMMs) is in general sound, but the employed methods require a revision. Detailed comments follow.

1. The review of the literature is rather poor. With respect to the empirical analysis, HMMs have been widely used in sport-data analysis and several extensions of the basic model are provided. Similarly, it is well-known that the two-step analysis leads to misleading inference; thus, dimensionality reduction and clustering should be performed simultaneously. See e.g. ROSTI, A. V. I. and GALES, M. J. F. (2002). Factor analysed hidden Markov models. In Proceedings of the IEEE International Conference on Acoustics, Speech and Signal Processing 949–952. YAO, K., PALIWAL, K. K. and LEE, T. W. (2005). Generative factor analyzed HMM for automatic

speech recognition. Speech Commun. 45 435–454. FIELD, M., STIRLING, D., PAN, Z. and NAGHDY, F. (2016). Learning trajectories for robot programming by demonstration using a coordinated mixture of factor analyzers. IEEE Trans. Cybern. 46 706–717. A. Maruotti. J. Bulla. F. Lagona. M. Picone. F. Martella. "Dynamic mixtures of factor analyzers to characterize multivariate air pollutant exposures." Ann. Appl. Stat. 11 (3) 1617 - 1648

2. The HMM model is not well defined. It is rather unclear how the linear predictor looks like and how the parameters were estimated. I guess the Gaussian distribution is considered, but no information about the variance is given. Moreover, no info about outliers are given. Model fitting and performance, residuals analysis, etc are not given. Indeed, neither the likelihood is specified. Overall, the manuscript lacks of formal definition of the modelling; thus, it cannot be accepted as the methods are not well introduced, described, etc. It is completely unclear if a multivariate model is considered or if PCs are analysed independently. Is the clustering obtained via local or global decoding?

3. The HMM assumes that the sojourn distribution is geometrically distributed. Please, provide evidence that this is plausible for the analysed data; extend the model to a flexible sojourn if the case.

4. I am wondering if 4 PCs are really needed and which differences arise if less or more PCs were considered.

5. Please, provide the code used to estimate the parameters, to ensure the reproducibility of the results, and more results of the software used.

Reviewer #2: This reviewer appreciates the the time and effort invested by the authors in reporting their study. I will present below some suggestions for revising the text and some questions about the research.

Specific comments:

1) The abstract states the existence of four main components, but only two are defined. I suggest defining the remaining two.

2) In the introduction to the article (lines 40-42), it is described: "Predictive models of individual athlete pacing in competitive kayak races and their change over a career can be used by coaches and sports scientists to better understand athlete progression and optimise strategies for peak performance". HMMs are also known for their use in predictive models. In their work, could HMMs predict changes in an athlete's pace during a race or throughout their career?

3) Was any form of data selection or exclusion applied (for example, incomplete sequences)?

4) I suggest presenting averages and deviations of the unnormalized speed data. I also suggest presenting statistical power.

5) (Line 137) Check for discrepancies between figure descriptions and their mention in the text.

6) (line 174) The term "fPC" appears in this line, but it is not described or defined earlier in the text. I suggest providing a brief explanation or definition of the "fPC" acronym when it is first introduced.

7) I suggest that the particular pacing characteristics or interpretations described for PC1, PC2, PC3 and PC4 be inserted into the text, in a clear and concise way, as soon as they are obtained.

8) (Line 196) State the meaning of 'AIC' in full, as it was not mentioned or defined earlier in the text. Is the number of states chosen for the research more closely related to the AIC (Akaike Information Criterion) than to other factors?

9) (Lines 255-258) How can the values described in Table 1, centroids for each state, be explained as being equal for women (K1 500m) and men (K1 1000m)?

10) (Lines 255-258)Table 1 presents all states (1 to 4) related to PCs (1 to 4). However, in Figure 5, there are PCs not linked to all states. Is this correct?

11) There appears to be a distinction between “states” and “predicted states”. At times, they seem to be synonyms, while in other instances they are not. I request that you observe this and suggest standardizing if necessary.

12) The text mentions 4 states, but in Figure 6, the scale for Athlete A (Predicted State) varies from 3 to 7 (with data variation in 4 states), and the scale for Athlete B varies from possibly 1 to 7 (with data variation in 5 states). In Figure 7, the scales for Athletes B and C vary from 1 to 4. Is this correct? Please explain.

13) (Lines 349-353) Is paragraph "Athlete A begins their career in State 2, displaying signs of a low dropoff/high kick pacing profile transitioning to State 4, average profile characteristics, when they move from U21s to U23s. Alternatively, Athlete B starts with a few races in State 3, low dropoff/high kick before transitioning to State 4 for the majority of their career. This appears to indicate that both of these athletes, who have the most pacing profiles in the data set, trends towards an average profile throughout their career." related to Figure 6? Verify the paragraph if it is indeed related to the figure or clarify the origin of the state values mentioned.

14) (Lines 356-358) Check the sentence regarding Figure 6:"Noticeably, this athlete is less consistent than Athlete A and B with several races identified as State 3, low dropoff/low kick, and therefore there kick appears to be inconsistent".

15) (Lines 367-368) "Therefore, it is possible that the male athletes will tend to be more consistent once they have a large data set in the Open age group." Is this conclusion based solely on Athletes C and D? Athlete D does not have data in the Open age group (Figure 7).

16) (Line 441) Check the formatting of reference number 11.

**Do you want your identity to be public for this peer review?** For information about this choice, including consent withdrawal, please see our Privacy Policy

Reviewer #1: No

Reviewer #2: No

---

## [Author Response · Author response to Decision Letter 1]

2 Oct 2024

We have attached a file contained all responses to the reviewers. All editor requirements have been fixed and a competing interests statement has been added to the cover letter.

---

## [Decision Letter · Decision Letter 1]

Dear Dr. Estreich,

Thank you for submitting your manuscript to PLOS ONE. After careful consideration, we feel that it has merit but does not fully meet PLOS ONE’s publication criteria as it currently stands. Therefore, we invite you to submit a revised version of the manuscript that addresses the points raised during the review process.

We look forward to receiving your revised manuscript.

Kind regards,

Matteo Vandoni

Academic Editor

PLOS ONE

Journal Requirements:

Additional Editor Comments:

Dear authors,

as you can see, revisor 1 asked to carefully revise some points..please provide a point by point response trying to asses his observations.

Kind regards

Reviewers' comments:

Reviewer's Responses to Questions

**Comments to the Author**

Reviewer #1: (No Response)

Reviewer #2: All comments have been addressed

2. Is the manuscript technically sound, and do the data support the conclusions?

Reviewer #1: Partly

Reviewer #2: Yes

3. Has the statistical analysis been performed appropriately and rigorously?

Reviewer #1: No

Reviewer #2: Yes

4. Have the authors made all data underlying the findings in their manuscript fully available?

Reviewer #1: No

Reviewer #2: Yes

5. Is the manuscript presented in an intelligible fashion and written in standard English?

Reviewer #1: Yes

Reviewer #2: Yes

Reviewer #1: Thank you very much for the efforts to reply to my comments.

Nevertheless, there are still several parts deserving clarifications and/or investigation.

1. As I mention before, the two-step analysis leads to misleading inference. Acknowledge this as a limitation of the study is not sufficient, as results may be unreliable if a joint approach is neglected. Moreover, it is rather unclear to me what the authors mean with "Our results have been shown to be robust in preliminary sensitivity analyses", more details on this are required.

2. I appreciate that more details on the HMM specification have been added to the main text. Are the Gaussian conditional densities with state-specific variances? If so, do you encounter any degeneracy issues? Moreover, please provide evidence that the Gaussian distribution is suited for the data at hand; the idea of removing outliers is questionable.

3. Residual analysis, qq-plot graphs, etc should be shown to ensure that the model is suitable for the data at hand. The AIC, and other model selection criteria, are useful to select the number of clusters (and to compare different model specifications) but not to guarantee that the model is adequate for the data at hand.

4. At last, one further point must be discussed and investigate, via models comparison. The HMM implicitly assume that the sojourn distribution is geometric. Please, check that this assumption is met and relax it if the case by assuming e.g. shifted negative binomial, logarithmic, etc sojourns. Please, provide evidence that the geometric sojourn is chosen according to any model selection criteria like the AIC.

Reviewer #2: I appreciate the authors for their responses to the queries raised. After a careful analysis, I am pleased to inform you that of the suggestions made in the previous review have been addressed. The changes implemented have improved the clarity and robustness of the work.

**Do you want your identity to be public for this peer review?** For information about this choice, including consent withdrawal, please see our Privacy Policy

Reviewer #1: No

Reviewer #2: No

---

## [Author Response · Author response to Decision Letter 2]

22 May 2025

I have attached a Response to Reviewers document that addresses all corrections to the paper

---

## [Editor Report · Decision Letter 2]

An analysis of pacing profiles in sprint kayak racing using functional principal components and Hidden Markov Models

PONE-D-24-05983R2

Dear Dr. Estreich,

We’re pleased to inform you that your manuscript has been judged scientifically suitable for publication and will be formally accepted for publication once it meets all outstanding technical requirements.

Kind regards,

Matteo Vandoni

Academic Editor

PLOS ONE
---

## [Editor Report · Acceptance letter]

PONE-D-24-05983R2

PLOS ONE

Dear Dr. Estreich,

I'm pleased to inform you that your manuscript has been deemed suitable for publication in PLOS ONE. Congratulations! Your manuscript is now being handed over to our production team.

Kind regards,

on behalf of

Dr. Matteo Vandoni

Academic Editor

PLOS ONE